# Impact of the COVID-19 Pandemic on the Diagnosis of Malignant Neoplasia of the Bronchus and Lung in the Burgos Region

**DOI:** 10.3390/healthcare12161677

**Published:** 2024-08-22

**Authors:** Gustavo Gutiérrez Herrero, Sandra Núñez-Rodríguez, Sergio Álvarez-Pardo, Jessica Fernández-Solana, Carla Collazo-Riobó, Álvaro García-Bustillo, Mirian Santamaría-Peláez, Jerónimo J. González-Bernal, Josefa González-Santos

**Affiliations:** 1Pulmonology Unit, Burgos University Hospital, 09006 Burgos, Spain; fggutierrez@ubu.es; 2Department of Health Sciences, University of Burgos, 09001 Burgos, Spain; jfsolana@ubu.es (J.F.-S.); ccollazo@ubu.es (C.C.-R.); agbustillo@ubu.es (Á.G.-B.); mspelaez@ubu.es (M.S.-P.); jejavier@ubu.es (J.J.G.-B.); mjgonzalez@ubu.es (J.G.-S.); 3Faculty of Health Science, University Isabel I, 09003 Burgos, Spain; sergio.alvarez@ui1.es

**Keywords:** non-communicable diseases, malignant lung neoplasia, COVID-19, healthcare, lifestyle, diagnosis, survival

## Abstract

Purpose: To retrospectively analyze the impact of the COVID-19 pandemic on the diagnosis, mortality rate, and survival period of malignant bronchial and lung neoplasms in the Burgos region, with the aim of promoting the development of strategies to improve cancer care management during health crises, highlighting the importance of non-pharmacological approaches to mitigate the negative impacts of future pandemics on lung cancer patients. Methods: This retrospective, longitudinal, single-center study was conducted in Burgos from 2019 to 2021. Participants included all patients diagnosed with malignant bronchial and lung neoplasms by the Pneumology unit of Complejo Asistencial Universitario de Burgos during the year immediately before and the year immediately after 31 March 2020, the official start date of the pandemic. Inclusion criteria encompassed patients diagnosed through histological or clinicoradiological methods, who provided informed consent. Data were systematically gathered using a specific template that included demographic information, disease stage, death, and survival time. Statistical analysis involved descriptive methods, ANOVA, and chi-square tests to assess differences in survival time and associations between categorical variables. Results: The results reveal a decrease in the number of patients diagnosed during the pandemic period (154 vs. 105), which could indicate delays in detection. However, there were no significant differences between the two periods, in which more than 60% of cases were detected in stage IV, being incompatible with survival. Although fewer patients died during the pandemic than expected (*p* = 0.015), patients diagnosed after the onset of the pandemic had a shorter survival time (182.43 ± 142.63 vs. 253.61 ± 224.30; *p* = 0.038). Specifically, those diagnosed in stage I during the pre-pandemic had a much longer survival time (741.50 days) than the rest of the patients (*p* < 0.05). In addition, among those diagnosed in stage IV, those diagnosed after the beginning of the pandemic had a shorter survival time (157.29 ± 202.36 vs. 241.18 ± 218.36; *p* = 0.026). Conclusions: Understanding these changes can support both medical strategies and non-pharmacological therapies to improve cancer care management during health crises, thus contributing to the optimization of public health.

## 1. Introduction 

Lung cancer is a leading non-communicable disease (NCD) that significantly contributes to global mortality [1]. The World Health Organization (WHO) estimates that NCDs, including cardiovascular diseases, cancer, respiratory diseases, and diabetes, account for over 41 million deaths annually, representing 71% of all premature deaths due to NCDs. [2]. Lung cancer in particular stands out due to its high incidence and mortality rate, posing a critical challenge to global public health [3,4,5,6]. The staging of lung cancer is crucial for determining the appropriate treatment and prognosis. Lung cancer is classified into stages I to IV, with stage I indicating localized cancer confined to the lungs and stage IV representing advanced cancer that has spread to other parts of the body. Early-stage detection (stages I and II) significantly improves the chances of successful treatment, while late-stage detection (stages III and IV) is associated with lower survival rates and more complex treatment requirements [7,8].

Lifestyle plays a crucial role in mitigating the risk of developing NCDs and improving the quality of life for those affected. In the context of lung cancer, adopting healthy habits, such as regular physical activity, a balanced diet, and reducing tobacco and alcohol consumption, has been shown to not only decrease the incidence of lung cancer, but also improve the prognosis, enhancing the efficacy of medical treatments [9,10,11,12]. Additionally, psychological and educational interventions are essential to support patients in managing their condition [13].

The COVID-19 pandemic, declared by the WHO on 11 March 2020 [14,15], profoundly impacted public health, with figures of over 649 million confirmed cases and more than 6.6 million deaths globally by December 2022 [16]. This global health crisis necessitated a massive reorganization of health services worldwide [17]. Although this reorganization was essential to address the health crisis, it created significant challenges and consequences on a global scale [18].

The suspension of routine medical services, the transformation of services to care for COVID-19, and containment measures significantly impacted not only COVID-19 patients, but also those with other medical conditions [18].

Malignant neoplasms are highly relevant in global health due to their high incidence and mortality rates, posing substantial challenges in early detection and comprehensive patient management [14,15,17,18,19,20,21,22,23,24,25].

In summary, according to similar studies, the number of new lung cancer diagnoses significantly decreased during the COVID-19 pandemic compared to pre-pandemic periods, and there was an increase in advanced-stage lung cancer diagnoses during the pandemic, with more patients presenting with stage IV disease. Consequently, the pandemic led to worse outcomes for lung cancer patients, with higher mortality rates and more severe disease at the time of diagnosis [26,27,28].

This study aims to retrospectively analyze the impact of the COVID-19 pandemic on the diagnosis, mortality rates, and survival period of malignant bronchus and lung neoplasm in the Burgos region. This analysis will develop strategies to improve cancer care management during a health crisis, emphasizing the importance of maintaining non-pharmacological approaches, such as promoting healthy lifestyles and psychological support, to mitigate the negative impact of future pandemics on the health of lung cancer patients [29]. 

## 2. Materials and Methods

### 2.1. Study Design 

This retrospective, longitudinal, single-center study was conducted in Burgos over the years 2019, 2020, and 2021. The primary objectives were to analyze the incidence of malignant neoplasms of the bronchus and lung and to determine if there were differences in survival time between patients diagnosed before and during the COVID-19 pandemic. Secondary objectives included analyzing the socio-demographic and clinical characteristics of diagnosed patients, such as age, gender, and disease stage, and determining if the disease stage was more advanced during the pandemic compared to prior periods.

### 2.2. Participants

The sample included all patients diagnosed with malignant lung and bronchial neoplasia by the Pneumology unit of the Complejo Asistencial Universitario de Burgos, Spain, during the year immediately prior to 31 March 2020 (the date on which the pandemic was officially declared) and the year immediately after that date. This two-stage sampling approach allowed us to capture the variability in incidence between the two periods.

Inclusion criteria for the study included patients diagnosed by histological or clinicoradiological methods.

Patients participated voluntarily, providing the necessary information for the collection of demographic data. Ethical and privacy principles were respected, and all participants gave informed consent for their inclusion in this study. The study adhered to ethical principles outlined in the Helsinki Declaration and was approved by the Clinical Research Ethics Committee of the University of Burgos with reference IO-06/2024 on 9 February 2024.

The exclusion criteria for this study comprised individuals who had not been diagnosed with cancer through histological or radiological methods and those who did not provide informed consent for the necessary data collection. These criteria ensured the reliability and ethical integrity of the collected data.

### 2.3. Procedure

Data collection was carried out systematically using a template specifically designed for this study. In this way, all relevant patient data were collected, ensuring the homogeneous collection of data and guaranteeing the consistency of the information collected. The template included the following information:Gender: Each patient was classified as male or female.Age: Ages were categorized into age ranges or intervals, to which patients were assigned. The age groups were defined as follows: <30 years, 30–39 years, 40–49 years, 50–59 years, 60–69 years, 70–79 years, 80–89 years, or ≥90 years.Disease stage: The stage of the disease was recorded for each patient at the time of diagnosis using the classification of stage I, II, III, or IV.Death: Patients who died because of the disease were followed up during the three years mentioned above, and the date of those who died because of the disease was recorded.Survival time: For deceased patients, the number of days of survival from the date of diagnosis to the date of death was calculated.

### 2.4. Statistical Analysis 

Statistical analysis was performed using SPSS version 28 statistical software (IBM-Inc., Chicago, IL, USA).

Descriptive methods were used, including the presentation of a table with the main clinical and socio-demographic data. Data were presented as number of cases and percentage of the total for categorical variables. Quantitative variables were subjected to the Kolmogorov–Smirnov normality test. Age followed a normal distribution, so it was presented as mean ± standard deviation. Survival time followed a non-normal distribution, so it was presented as median (interquartile range).

As data were collected before and after the onset of the COVID-19 pandemic, several comparisons were made between the main study variables.

Although the data for the survival time variable were not normally distributed, it was decided to perform parametric tests on those analyses that included this variable. Some studies have shown empirical evidence of the robustness of the ANOVA test to other non-parametric analyses, even in contexts involving non-normal distributions [30].

Therefore, to analyze whether there were statistically significant differences in the number of days of survival between the two periods, parametric ANOVA tests were used.

In addition, chi-square tests were also carried out to assess the possible associations between the different categorical variables. To determine whether there were significant differences between expected frequencies and observed frequencies, absolute values greater than 1.96 or −1.96 in the corrected residuals were considered.

## 3. Results

### 3.1. Sociodemographic and Clinical Characteristics of the Sample

Table 1 shows the main socio-demographic and clinical characteristics of the patients included in this study.

The total study sample consisted of 259 patients, of whom 154 were diagnosed before the pandemic and 105 after the declaration of the pandemic. Of these, 203 (78.4%) were men and 56 (21.6%) were women. As can be seen in Table 1, males represented a much higher percentage both pre-pandemic and during the pandemic.

The total mean age of the diagnosed patients was 68.77 ± 8.89 years. During these two periods, no cases of malignant neoplasm of the bronchus and lung were diagnosed in persons younger than 40 years or older than 89 years. In both periods, most cases were diagnosed in patients aged 60–79 years.

Almost two thirds (62.9%) of the total cases diagnosed during these periods were already in stage IV, 21.6% in stage III, 6.2% in stage II, and 9.3% still in stage I.

Of the total number of patients diagnosed, 105 (40.5%) were still surviving at the time of the end of follow-up of this study, while 154 (59.5%) died during follow-up. Among all of these deceased patients, the median number of days from diagnosis to death was 181.0 (66.0–332.0) days.

### 3.2. Association between Pandemic and Age, Gender, Cancer Stage, and Mortality

Table 2 below shows the inferential results obtained after associating the age groups with the two time periods analyzed. As can be seen, the *p*-value is greater than 0.05, which means that there were no statistically significant differences between the expected frequencies and the observed frequencies according to the age of the patients.

Table 2 also shows the inferential results obtained after associating the gender of the patients with the time periods analyzed. As can be seen, the *p*-value is also greater than 0.05, which means that there were also no statistically significant differences between the expected frequencies and the observed frequencies according to the age of the patients diagnosed with malignant neoplasm of the bronchus and lung.

It also shows the inferential results obtained after associating cancer stage with the two periods analyzed. As can be seen, the *p*-value is greater than 0.05, which means that there were also no statistically significant differences between the expected and the observed frequencies according to the stage.

Finally, Table 2 also shows the inferential results obtained after associating mortality with the two time periods analyzed. As can be seen, there was a statistically significant association (*p* = 0.015) between the two variables. After analyzing the corrected residuals, the number of patients who died during the pre-pandemic period was higher than expected, and, on the contrary, it was lower than expected during the pandemic period.

### 3.3. Association between Pandemic, Cancer Stage, and Years of Survival

Table 3 below shows the differences in survival years between the two years analyzed. As can be seen, there were statistically significant differences between the two groups (*p* = 0.038). The mean number of days of survival was higher during the pre-pandemic period than during the pandemic.

Table 3 also shows the differences in survival years between the two years analyzed, considering the different stages of the disease.

As can be seen, there were statistically significant differences between the groups (*p* = 0.004). After analyzing multiple comparisons between the different groups, statistically significant differences (*p* < 0.05) were found between those patients diagnosed with stage I cancer before the pandemic versus the rest of the patients, regardless of their stage of disease and whether they were diagnosed before or after the pandemic. The pre-pandemic stage I diagnosis has a much longer survival period compared to the others.

In addition, there were also significant differences (*p* = 0.026) between those patients diagnosed with stage IV cancer before and after the pandemic. Pre-pandemic stage IV diagnosis showed a longer survival time compared to stage IV diagnosis in the pandemic period.

## 4. Discussion

Malignant neoplasm of the bronchus and lung poses a significant global challenge due to its high incidence and mortality rates [3,4,5,6]. Patients with this disease are particularly vulnerable to developing COVID-19 and related complications [17,18,19,31]. Furthermore, the pandemic has burdened the healthcare system, likely causing delays in lung cancer diagnosis [17,18,32,33]. Systemic changes in healthcare access and delivery during the COVID-19 pandemic likely influenced diagnostic practices for malignant neoplasms of the bronchus and lung. The redirection of healthcare resources towards managing COVID-19 patients, coupled with the implementation of infection control measures and lockdowns, potentially led to reduced accessibility to diagnostic services for non-COVID-19 conditions. This could have resulted in delays in routine screenings, diagnostic procedures, and follow-up appointments for suspected cases of lung cancer. Moreover, patient hesitancy or fear of contracting COVID-19 in healthcare settings might have deterred individuals from seeking timely medical evaluation. These systemic disruptions may have contributed to the observed decrease in the number of diagnosed cases during the pandemic period and could have exacerbated the presentation of cases at more advanced stages, underscoring the need for adaptive strategies in healthcare delivery to maintain diagnostic vigilance even during public health crises [17,18,32,33]. This is a clear issue, as early disease detection is essential; disease progression can lead to fatal outcomes [20,21]. Moreover, understanding the diagnosis and disease stage is vital for providing necessary care to manage symptoms and improve the quality of life of affected patients. Therefore, the aim of this study was to analyze the impact of the COVID-19 pandemic on the diagnosis, mortality rates, and survival time in patients diagnosed with this disease in the Burgos region.

Similar to other studies, the results of our study revealed a concerning decrease in the number of patients diagnosed during the pandemic period [21,23]. This reduction could be attributed to several factors, including healthcare system overload, resource redistribution, and reduction in various services [22,23,24]. These factors combined may have contributed to delays in lung cancer detection, posing a significant challenge, as late detection is associated with lower survival likelihoods [20,21]. In our study, the majority of cases were diagnosed at advanced stages, both in the pre-pandemic and pandemic periods. These findings underscore the importance of learning from this crisis to improve early detection and effective disease management in the future [34].

It is crucial to recognize that lung cancer is not only treated with medical approaches, but also requires a comprehensive approach, including non-pharmacological therapies and lifestyle changes [9,10,11,12]. While medical treatments are vital, adopting healthy habits such as smoking cessation, maintaining a balanced diet, and regular physical exercise can play a crucial role in disease management [10]. However, during the COVID-19 pandemic, maintaining these healthy habits has become even more challenging due to movement restrictions, emotional stress, and disruption of support services [35,36,37]. These factors may have contributed to worsened outcomes in lung cancer patients, particularly those diagnosed at advanced stages.

Contrary to expectations, a higher mortality rate was observed during the pre-pandemic period compared to the pandemic period. These data likely do not fully reflect reality, and are justified by the fact that patients diagnosed during the pandemic had a shorter longitudinal follow-up period. It is likely that the disease progressed in many of them, and death occurred after the end of the study follow-up. However, studies like the one of Addabbo et al. also show that there is no significant difference in lung cancer deaths before and during the pandemic, attributing these results to effective measures taken [38].

In terms of survival time, the results showed that patients diagnosed during the pandemic had a shorter survival time from diagnosis to the date of death. Specifically, those patients diagnosed with cancer at an early stage (I) before the pandemic had much longer survival times than other patients, regardless of their disease stage and whether they were diagnosed before or after the pandemic. This phenomenon highlights the dual effect of late detection and diagnosis during the pandemic, resulting in compromised outcomes for patients [39].

Furthermore, those patients diagnosed with cancer at an advanced stage (IV) after the pandemic also demonstrated shorter survival times compared to those at similar stages of the disease. This suggests that, despite inherently low survival prospects in advanced stages, the challenges posed by the COVID-19 health crisis likely contributed to delays in diagnosis and complexities in patient management, as well as in active disease management, accelerating their demise. These findings emphasize the multifaceted impact of the pandemic on cancer care, including delays in diagnosis, alterations in treatment protocols, and challenges in managing advanced stages of the disease, all of which culminate in worse outcomes for patients.

One of the main limitations of this study was the small sample size, primarily due to its single-center nature. This limitation restricted the generalization of findings to broader populations. Additionally, the retrospective design of the study introduced inherent limitations, including possible biases and limitations in the availability and quality of data. Furthermore, being a single-center study, the findings may not fully capture the diversity of experiences and practices observed in broader contexts. These limitations should be considered when interpreting the results and extrapolating them to other settings or populations. Despite the identified limitations, this study provides valuable information on the impact of the COVID-19 pandemic on the diagnosis, mortality rates, and survival time of patients with malignant bronchus and lung neoplasms in the Burgos region. By shedding light on the challenges faced by healthcare systems and patients during this unprecedented crisis, the findings underscore the urgent need for proactive measures to mitigate adverse effects on cancer diagnosis and treatment, as well as to create effective protocols for oncological diagnosis and treatment during health crisis situations. Additionally, this study highlights the critical importance of early detection and timely intervention to improve patient outcomes and reduce mortality rates associated with lung cancer. As the global community continues to navigate the complexities of the COVID-19 pandemic and its implications for healthcare delivery, the findings presented in this article serve as a valuable resource to inform evidence-based strategies aimed at optimizing cancer care in the face of future public health emergencies.

Recognizing a limitation of this study’s single-center nature, as a future line of research, we propose increasing the study sample, including multiple centers from various regions, to generalize the obtained results to the entire population.

## 5. Conclusions

The main findings of this study shed light on several critical aspects concerning the impact of the COVID-19 pandemic on the diagnosis and survival outcomes of patients with malignant neoplasm of the bronchus and lung. Firstly, the observed decrease in diagnosed cases during the pandemic underscores the significant strain placed on the healthcare system, resulting in delays and disruptions in routine care pathways. Moreover, this study reveals a substantial reduction in survival time among patients diagnosed during the pandemic, particularly evident in stages I and IV of the disease.

These findings underscore the urgent need for further investigations into the underlying factors contributing to delayed diagnosis and compromised survival outcomes during health crises such as the COVID-19 pandemic. By elucidating these factors, future research can inform the development of targeted interventions and policies aimed at improving cancer care delivery and optimizing patient outcomes during times of crisis. By elucidating these factors, future research can inform the development of specific interventions and policies aimed at improving cancer care delivery and optimizing patient outcomes during times of crisis. Firstly, it is essential to strengthen diagnostic pathways by implementing more robust and flexible systems, such as telemedicine and remote diagnostics, to facilitate early detection and reduce delays. Additionally, protocols should be established to ensure that cancer patients receive priority care during medical emergencies. Understanding and applying these changes can significantly enhance cancer care management during medical crises, optimizing patient outcomes and strengthening the resilience of the healthcare system.

## Figures and Tables

**Table 1 healthcare-12-01677-t001:** Sociodemographic and clinical characteristics of the sample.

	Pre-Pandemic	Pandemic
Diagnosed patients, *n*	154	105
Gender		
Male, *n* (%)	122 (79.2)	81 (77.1)
Female, *n* (%)	32 (20.8)	24 (22.9)
Age, Mean ± SD	68.57 ± 8.99	69.06 ± 8.77
40–49 years, *n* (%)	2 (1.3)	1 (1.0)
50–59 years, *n* (%)	24 (15.6)	14 (13.3)
60–69 years, *n* (%)	58 (37.7)	35 (33.3)
70–79 years, *n* (%)	50 (32.4)	44 (41.9)
80–89 years, *n* (%)	20 (13.0)	11 (10.5)
Stage		
I, *n* (%)	13 (8.5)	11 (10.5)
II, *n* (%)	12 (7.8)	4 (3.8)
III, *n* (%)	35 (22.7)	21 (20.0)
IV, *n* (%)	94 (61.0)	69 (65.7)
Decease		
No, *n* (%)	53 (34.4)	52 (49.5)
Yes, *n* (%)	101 (65.6)	53 (50.5)
Number of days of survival since diagnosis, Median (IQR)	193.0 (68.5–376.5)	160.0 (58.5–298.5)

SD = standard deviation; IQR = interquartile range.

**Table 2 healthcare-12-01677-t002:** Chi-square test to determine the association between pandemic and age, gender, cancer stage, and mortality.

		Pre-Pandemic	Pandemic
40–49	Count	2	1
	Expected Count	1.8	1.2
	Corrected residual	0.3	−0.3
50–59	Count	24	14
	Expected Count	22.6	15.4
	Corrected residual	0.5	−0.5
60–69	Count	58	35
	Expected Count	55.3	37.7
	Corrected residual	0.7	−0.7
70–79	Count	50	44
	Expected Count	55.9	38.1
	Corrected residual	−1.6	1.6
80–89	Count	20	11
	Expected Count	18.4	12.6
	Corrected residual	0.6	−0.6
Male	Count	122	81
	Expected Count	120.7	82.3
	Corrected residual	0.4	−0.4
Female	Count	32	24
	Expected Count	33.3	22.7
	Corrected residual	−0.4	0.4
Stage I	Count	13	11
	Expected Count	14.3	9.7
	Corrected residual	−0.6	0.6
Stage II	Count	12	4
	Expected Count	9.5	6.5
	Corrected residual	1.3	−1.3
Stage III	Count	35	21
	Expected Count	33.3	22.7
	Corrected residual	0.5	−0.5
Stage IV	Count	94	69
	Expected Count	96.9	66.1
	Corrected residual	−0.8	0.8
Non−deceased patient	Count	53	52
	Expected Count	62.4	42.6
	Corrected residual	−2.4	2.4
Deceased patient	Count	101	53
	Expected Count	91.6	62.4
	Corrected residual	2.4	−2.4

Pandemic and age: X^2^ (259) = 2.467, *p* = 0.651. Pandemic and gender: X^2^ (259) = 0.159, *p* = 0.690. Pandemic and cancer stage: X^2^ (259) = 2.314, *p* = 0.510. Pandemic and mortality: X^2^ (259) = 5.912, *p* = 0.015.

**Table 3 healthcare-12-01677-t003:** ANOVA test to compare years of survival time between pre-pandemic and pandemic diagnoses, considering the different cancer stages.

	Number of Days of Survival since Diagnosis		ANOVA	
	Mean	Standard Deviation	F	Sig. (*p*)
Pre-Pandemic	253.61	224.30	4.396	0.038
Pandemic	182.43	142.63		
Pre-Pandemic Stage I	741.50	228.40	3.191	0.004
Pre-Pandemic Stage II	212.89	215.33		
Pre-Pandemic Stage III	271.18	204.27		
Pre-Pandemic Stage IV	241.18	218.36		
Pandemic Stage I	329.50	73.54		
Pandemic Stage II	195.00	148.81		
Pandemic Stage III	287.71	132.13		
Pandemic Stage IV	157.29	202.36		

*p* (pre-pandemic stage I vs. all other groups) < 0.05. *p* (pre-pandemic stage IV vs. pandemic stage IV) = 0.026.

## Data Availability

Data are contained within the article.

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
