# Peer review of "Impact of the COVID-19 Pandemic on the Diagnosis of Malignant Neoplasia of the Bronchus and Lung in the Burgos Region"

_healthcare, 2024, doi:10.3390/healthcare12161677_

Round 1

Reviewer 1 Report

Comments and Suggestions for Authors

Dear author, thank you very much for reviewing your manuscript. I give you the following comment to address in your manuscripts. I recommend that you please add figures regarding your results.

Best regards

Major Comments:

  1. Purpose and Methods Clarity: Check that the abstract explains the main goal and the procedures used in detail, including the standards for patient inclusion and data-collecting procedures.
  2. Comparison Metrics: Give more info on the specific metrics utilised to compare the pre-pandemic and pandemic periods, especially about survival rates, disease stage distribution, and demographics.
  3. Statistical Analysis: Please provide more details on the statistical techniques (such as ANOVA and chi-square tests), their applicability to the data, and how they influenced the results.
  4. Analysis of Survival: Provide additional details on the findings of the survival analysis, including how the survival times should be interpreted and how this may affect clinical practice.
  5. Implications for Medical Strategy and Policy: Describe the various ways the results may influence medical practices and regulations, including concrete suggestions for enhancing cancer treatment in times of medical emergency.

Minor Comments:

  1. Language Precision: Improve the precision and clarity of technical terminology and results so researchers and medical professionals can understand them.
  2. Data Representation: For easier comparison and interpretation, think about presenting significant numerical results (such as survival times and mortality rates) more consistently.
  3. Description of the Cohort: To put into perspective the study population and its significance to broader healthcare issues in context, including a few lines on patient demographics (e.g., age, gender distribution).
  4. Survival Time Interpretation: Give an overview of the calculation methods used to determine survival times and any biases or confounding variables that may have affected the results' interpretation.
  5. Future Research Directions: Describe possible avenues for future research, such as examining obstacles to early diagnosis or evaluating telemedicine's efficacy in maintaining cancer care during pandemics.

Reviewer 2 Report

Comments and Suggestions for Authors

Thank you for submitting your manuscript titled Impact of the COVID-19 pandemic on the diagnosis of malignant neoplasia of the bronchus and lung in the Burgos region” for consideration. I appreciate the opportunity to review this work, which addresses the important topic of how the COVID-19 pandemic has affected lung cancer diagnosis and management.

I want to acknowledge that I have previously reviewed a very similar version of this study. After careful consideration of this current manuscript, I find that my initial concerns still apply. While your topic is relevant, there are several aspects of the study that require significant improvement to increase its validity and impact:

1.      The single-centre design and focus on a specific region limit the generalizability of your findings. Consider expanding your study to include multiple centres across different regions.

2.      The sample size remains limited, which affects the statistical power of your analysis. A larger sample would improve the reliability of your findings.

3.      The retrospective nature of the study introduces potential biases. Consider applying more advanced statistical analyses (e.g., multivariable regression analysis, Cox proportional hazards model, propensity score matching) to control for confounding variables.

4.      The study would also benefit from a deeper assessment of how systemic changes in healthcare access and delivery during the pandemic may have influenced diagnostic practices.

5.      The theoretical framework could be strengthened by situating your findings within the broader global context and comparing them to similar studies.

I hope these comments are helpful in refining your work. Thank you for your contribution to this important field of study.

Reviewer 3 Report

Comments and Suggestions for Authors

The article discusses an interesting topic that sheds light on health challenges during the COVID-19 pandemic. It not only focuses on issues directly related to the virus but also considers pre-existing pathologies. Researchers at the medical center explore this perspective, acknowledging its limitations while recognizing the opportunity for large-scale studies.  It’s crucial for researchers to address disease stage classification in their work. However, the introduction and methods sections do not currently cover this aspect. To enhance understanding, consider reducing the number of tables and providing clearer explanations of the results. Specifically, focus on tables where the same statistical analysis is used.   Specific comments: There should be a space after (16) on line 50. Remove the extra parenthesis after (26) on line 133. In Table 1, the sum of the percentage of ages in the “pre-pandemic” group results in 100.1%—this should be corrected. Maintain consistency: Use either 0.05 or .05 throughout the document.

Round 2

Reviewer 2 Report

Comments and Suggestions for Authors

Thank you for submitting the revised version of your manuscript titled “Impact of the COVID-19 pandemic on the diagnosis of malignant neoplasia of the bronchus and lung in the Burgos region”. I appreciate the opportunity to review this work again and acknowledge your efforts to address the concerns raised in my initial review.

I want to commend you on the improvements made to the manuscript, particularly:

1.      The addition of a useful discussion on systemic healthcare changes during the pandemic and their potential impact on diagnostic practices for lung cancer. This provides important context for interpreting your results.

2.      The inclusion of a summary of findings from similar studies in the introduction, which strengthens the theoretical framework and better situates your research within the existing literature.

However, after careful consideration of the revised manuscript and your responses, I regret to say that several fundamental methodological limitations persist. These issues continue to affect the overall validity and impact of the study:

1.      While you acknowledge the single-centre design as a limitation and mention it as a future line of research, the current study remains limited to one region. This significantly restricts the generalizability of your findings to broader populations or healthcare systems.

2.      Although you recognize the small sample size as a main limitation, it remains unaddressed in the current revision. The limited number of patients continues to constrain the statistical power of your analysis, making it difficult to draw robust conclusions.

3.      Your explanation of the choice of statistical methods based on data limitations and comparability with other studies is appreciated. However, this does not fully address the concern about adequately controlling for potential confounding variables, which is crucial for the validity of your conclusions.

Given these persistent limitations, I believe that addressing these core methodological issues would require substantial revisions beyond what has been accomplished in this revision. These changes are necessary to improve the validity and applicability of your findings.

I encourage you to pursue your mentioned future lines of research, particularly: a) expanding to a multi-centre approach to increase sample size and improve generalizability, and b) implementing more advanced statistical analyses to control for confounding variables, with a larger dataset if possible. With these substantial revisions, your research could make a more significant contribution to the field.

I hope these comments are helpful in guiding your future research efforts.

Reviewer 3 Report

Comments and Suggestions for Authors

The work presented by the researchers meets the requirements outlined in the previous review. They have significantly improved the compilation of tables in the results section and made the requested adjustments to the percentages.

Additionally, they included a classification system for a more thorough description of the type of cancer studied, enhancing the overall understanding of the topic.

Thank you for considering my observations
